# Influence of Clinical and Psychosocial Factors on the Adherence to Topical Treatment in Psoriasis

**DOI:** 10.3390/healthcare12080822

**Published:** 2024-04-12

**Authors:** Ana Teixeira, Maribel Teixeira, Rita Gaio, Tiago Torres, Sofia Magina, Maria Alzira Pimenta Dinis, José Sousa-Lobo, Isabel Almeida, Miguel Peixoto, Vera Almeida

**Affiliations:** 1Associate Laboratory i4HB—Institute for Health and Bioeconomy, University Institute of Health Sciences—CESPU, 4585-116 Gandra, Portugal; ana.teixeira@iucs.cespu.pt; 2UCIBIO—Applied Molecular Biosciences Unit, Translational Toxicology Research Laboratory, University Institute of Health Sciences (1H-TOXRUN, IUCS-CESPU), 4585-116 Gandra, Portugal; 3Centro de Matemática da Universidade do Porto, Departamento de Matemática, Faculdade de Ciências, Universidade do Porto, 4169-007 Porto, Portugal; argaio@fc.up.pt; 4Serviço de Dermatologia, Centro Hospitalar do Porto, Hospital de Santo António, 4099-001 Porto, Portugal; torres.tiago@outlook.com; 5Serviço de Dermatologia, Centro Hospitalar de São João, Departamento de Farmacologia e Terapêutica, Faculdade de Medicina, Universidade do Porto, 4200-319 Porto, Portugal; smagina@med.up.pt; 6Fernando Pessoa Research, Innovation and Development Institute (FP-I3ID), University Fernando Pessoa (UFP), Praça 9 de Abril 349, 4249-004 Porto, Portugal; madinis@ufp.edu.pt; 7UCIBIO—Applied Molecular Biosciences Unit, MedTech, Applied Biomolecular Biosciences Unit, Medicines and Healthcare Products, Faculdade de Farmácia, Universidade do Porto, 4050-313 Porto, Portugal; slobo@ff.up.pt (J.S.-L.); vera.almeida@iucs.cespu.pt (V.A.); 8Instituto Universitário de Ciências da Saúde (IUCS), CESPU—Cooperativa de Ensino Superior Politécnico e Universitário, 4585-116 Gandra, Portugal; mikapeixoto@gmail.com; 9Psychosocial Rehabilitation Laboratory, Rehabilitation Investigation Center, School of Health, Polytechnic University of Porto, 4200-465 Porto, Portugal; 10UNIPRO—Unidade de Investigação em Patologia e Reabilitação Oral, Instituto Universitário de Ciências da Saúde (IUCS), CESPU—Cooperativa de Ensino Superior Politécnico e Universitário, 4585-116 Gandra, Portugal

**Keywords:** Brief Symptoms Inventory instrument (BSI), clinical and psychosocial factors, psoriasis, medication log (med log), topical treatment, treatment adherence

## Abstract

(1) Background: Psoriasis is a common chronic inflammatory skin disease with different manifestations, affecting the quality of life at social, emotional, and professional dimensions and requiring long-term treatment. This study aimed to investigate the effect of psychosocial and clinical factors on adherence to topical treatment in psoriasis. (2) Methods: Self-reported measures and weighing the medicines were used to assess adherence. Psychopathological symptoms were measured using the Brief Symptoms Inventory (BSI). Social and clinical factors were assessed by a sociodemographic and clinical questionnaire. Adherence to treatment with topical medication was assessed using a sample of 102 psoriasis patients. (3) Results: The explanatory models of adherence to topical treatment in psoriasis translated into positive associations between adherence and the education level (higher education) (*p* = 0.03; φ = 0.23), the single-family household (*p* = 0.01; φ = 0.44), active employment status (*p* = 0.05; φ = −0.19), familiar history of psoriasis (*p* = 0.04; φ = −0.21), and the presence of obsessive-compulsive symptoms (*p* = 0.01; *d* = 0.29). (4) Conclusions: In patients who present the characteristics identified that influence non-adherence, instructions should be reinforced to increase adherence. The experimental mortality (39.6%) reduced the sample size, representing a limitation of the study.

## 1. Introduction

Psoriasis is an inflammatory skin disease, diagnosed by the characteristic well-defined raised erythematous psoriatic plaques, with silvery white scales, localized preferentially on the extensor surfaces [1]. The redness and scaliness of the lesions, itching, and pain symptoms associated with psoriasis affect the daily activities and social relationships of patients, particularly when involving body exposure [2,3]. Due to the body image change resulting from the disease, the patients may feel low self-esteem and stigmatization, particularly when lesions are located in a visible part of the body [4]. Contrary to this, patients who have achieved a remission of psoriasis lesions showed that psoriasis had almost no impact on their daily lives [5]. Psoriasis is a chronic dermatological condition often accompanied by psychopathological symptoms, potentially involving demanding treatment regimens affecting daily organization [6]. Adequate psoriasis treatment depends on its clinical type and severity, the patient’s preference, i.e., cost and convenience, and the impact of this dermatosis on the patient’s quality of life [7,8]. Furthermore, being female, unemployment status, and smoking (both current and former) are variables that hinder treatment effectiveness [9]. Two main different psoriasis treatment types are available: (i) topical treatments, recommended for patients with mild to moderate psoriasis and for newly diagnosed patients, include corticosteroids, vitamin D derivatives, tazarotene, anthralin, tacrolimus, pimecrolimus, and newer tar formulations; (ii) non-topical treatments, for patients with more severe forms of psoriasis, include phototherapy and systemic medicines, i.e., conventional or biologic agents [10,11,12]. When it comes to treatment, it has been highlighted that some misdiagnoses for psoriasis severity can lead to undertreatment, and the International Psoriasis Council suggests that instead of dividing patients according to severity, patients should be divided into candidates for topical treatment or candidates for systemic therapy (which includes biological and non-biological treatments) according to a set of criteria [13]. Although topical medicines are usually the first-line treatment for mild and moderate psoriasis, low adherence restricts the clinical success of this therapeutic strategy with low side effects. Treatment adherence is the extent to which patients’ treatment-related behaviors agree with a health professional’s advice. In long-term chronic diseases, e.g., psoriasis, therapy adherence is generally poor, decreasing over time, with non-adherence leading to suboptimal health outcomes, compromising the quality of life and increasing healthcare costs [14]. Specifically for psoriasis, increased disease severity [15] and decreased quality of life [16] are associated with lower adherence. For psoriasis topical treatment in particular, low adherence values have been reported, ranging from 39% to 73% [17,18,19], with varying degrees of adherence across different treatments [20]. In a review comprising 20 studies in relation to assessment methods of psoriasis topical treatment adherence, adherence was highly variable [21]. In order to improve the quality and reliability of treatment adherence studies, the combination of different assessment methods has been suggested. This approach included the use of self-reported methods (e.g., adherence questionnaire and med log) and a method that measures the amount of the applied medicine [21,22]. Using this approach, the authors of this study have studied the influence of the mechanical properties of the pharmaceutical dosage form on adherence to topical treatment in psoriasis [23]. Specific personality traits such as extraversion, openness to new experiences, kindness, and conscientiousness can influence treatment adherence [24]. Psychological support can improve adherence to topical treatment and the quality of life in patients with psoriasis [25,26,27]. Considering the importance of the biopsychosocial dimension in disease management, the treatment of psoriasis should be the responsibility of a multidisciplinary team of health professionals, e.g., physicians, nurses, pharmacists, and psychologists [28]. Quality of life and psychological factors, e.g., psychological distress and patient satisfaction with treatment, are associated with adherence. Adherence is positively associated with treatment satisfaction and negatively associated with psychological distress [16,29]. Given the correlation between treatment satisfaction and adherence, factors such as convenience, safety, treatment type, perceived quality of life improvements, and response time, which are positive predictors of treatment satisfaction [30], could indirectly predict treatment adherence. Patients with psoriasis have a high prevalence of several mental disorders [31,32]. Some studies have examined the reasons for psoriasis treatment adherence in patients. Some have centred on psychosocial issues to justify treatment adherence. However, only a few have focused on the role of psychological and clinical variables, together, in adherence to treatment exclusively with topical medicines. Aiming to fill this gap, this study analyses the influence of the psychosocial and clinical factors on psoriasis topical treatment adherence.

## 2. Materials and Methods

### 2.1. Participants

The 102 patients engaged in this study were recruited from Portuguese private and public health institutions, familial health unities, hospitals, and a psoriatic patients’ association (PSOPortugal). After accepting to participate in the study, each participant was contacted after a physician consultation where the study‘s scope was explained. Additionally, some volunteers also interested in participating contacted the researchers via telephone or email. The following criteria were used to select the participants: over 18 years old, being treated exclusively with topical medicines (gels, creams, or ointments), and with a physician’s prescription for the psoriasis treatment. Only one medicine was chosen for each patient in the adherence study, aiming to simplify the patient’s participation and reduce the possibility of registration errors. Exclusion criteria were pregnancy, phototherapy or systemic treatment during the assessment period, illiteracy, and severe psychiatric comorbidity. All participants had to sign an informed consent. The study was previously approved by the Ethics Committee of the University of Porto; IUCS (Instituto Universitário de Ciências da Saúde/CESPU, ARS (Administração Regional de Saúde) Norte, reference number 70, study T421; Hospital de São João; Hospital de Santo António; CNPD (Comissão Nacional de Proteção de Dados), process number 1463/2014, authorization 5343/2014.

### 2.2. Measures

#### 2.2.1. Sociodemographic and Clinical Questionnaire

The demographic and clinical information of patients was obtained relating to age, gender, marital status, education, professional status, and the family unit. Clinical issues were also assessed, e.g., psoriasis family history, disease duration, daily life changes due to psoriasis (measured in number of days), work days of absence per year, comorbidity, anxiolytic or antidepressant medication, and a social impact of psoriasis.

#### 2.2.2. Severity Assessment

The Portuguese version of the simplified and self-administered psoriasis area and severity index (SAPASI-PT) was used for the patient’s severity self-assessment. On a visual analogic scale, patients were asked to shade the location of their psoriasis lesions on the front and back of a human figure and indicate their perception of three features of lesions—colour, thickness, and scaliness—based on a visual analogic scale (VAS). The investigator assessed the affected area to rate the instrument, which was then used to calculate the body surface area for each of the following four areas: head, upper extremities, trunk, and lower extremities. The fraction of the total surface area affected was graded on a 0–6 scale (0 for no involvement; up to 6 for greater than 90% involvement). The SAPASI score was calculated using the affected area’s data, as well as the three features of the lesions listed above. A score of zero indicated remission of the disease; a score between 0 and 3 indicated a mild form of psoriasis; a score between 3 and 15 indicated a moderate form of psoriasis; and a score higher than 15 indicated a severe form of the disease [2].

#### 2.2.3. Brief Symptoms Inventory (BSI)

BSI assesses psychopathological symptoms by nine dimensions of symptomatology, i.e., somatization, obsession/compulsion, interpersonal sensitivity, depression, anxiety, hostility, phobic anxiety, paranoid ideation, and psychoticism, and three global indices, i.e., global severity index (GSI), positive symptom total (PST), and positive symptom distress index (PSDI). BSI includes 53 question items, which must be answered on a 5-point Likert scale, ranging from 0 (not at all) to 4 (extremely). Dimension scores are calculated by summing the values for the items included in that dimension and dividing by the number of items endorsed in that dimension. Derogatis [33] found Cronbach’s alpha values between 0.71 for psychoticism and 0.85 for depression. Canavarro [34] found Cronbach’s alpha values between 0.72 for psychoticism and the same 0.85 for depression. In this study, Cronbach’s alpha values between 0.64 for paranoid ideation and 0.80 for somatization were found.

#### 2.2.4. Adherence Measurement

Adherence to topical treatment was assessed through self-reported measurements including a Questionnaire for Adherence to TOPical Treatment (QATOP) [16] and a med log. A QATOP was used for the identification of reasons for non-adherence and treatment-associated variables, and the med log for the registration of the administration frequency. A third method of adherence measurement was the use of the medication weight for the assessment of the administrated dose. At the beginning and the end of the study (45 days), the medicine packages were weighed and adherence by medication weight was calculated using Equation (1).
*W* = *W*u/*W*ex × 100(1)
where *W* represents the medication weight adherence (%), *W*u corresponds to the medication weight used per application = (weight dispensed − weight returned)/number of applications registered in the medication log, and *W*ex represents the expected medication weight per application = 0.25 × body surface area (BSA). Values of BSA were obtained using SAPASI-PT. From the average of adherence evaluated by the med log and medication weight, adherence to medication, named the adherence combo, was determined according to the equation [(adherence med log + adherence medication weight)/2]. To avoid overestimation of adherence values, for values higher than 100%, the result was subtracted from 200 (e.g., 120% value is converted to 80%). To study the effect of psychosocial factors on adherence to topical treatment, the sample was divided into two different groups, based on the adherence combo: (1) “adherent” (with adherence values of 80–120%) and (2) “non-adherent” (values ˂ 80% or >120%) to the prescribed treatment, with relative frequencies of 19.8% and 80.2%, respectively, in groups 1 and 2 [35]. These results allow us to infer the high frequency of individuals who do not adhere to treatment with topical medicines. A consensual standard for identification of individuals adhering to treatment does not exist. However, most studies consider a maximum deviation of ±20%, a criterion that was followed in the study of Jevtić, Bukumirić and Janković [36].

### 2.3. Procedures

The study protocol was applied to a sample of 102 psoriasis patients, in a longitudinal design, at two subsequent moments. It started after a medical consultation and opening of a new medicine package; after approximately 45 days, 67 patients (39.6%) recruited at the first moment did not complete the study protocol. This period was considered by the experts’ panel, composed of dermatologists, pharmacists, psychologists, and statistics, as the appropriate to study psoriasis topical treatment adherence since it is a period needed to evaluate the clinical effectiveness of the treatment. At the first assessment moment, the clinical and sociodemographic questionnaire, BSI, and SAPASI were filled out and the med log was delivered. Patients were asked to fill in the med log concerning the number of daily applications of the topical treatment. At the second assessment moment, QATOP and the SAPASI were applied and the med log was collected. The medicines used by the patients were weighed.

### 2.4. Statistical Analyses

Descriptive statistics were used to characterize the sample’s sociodemographic and clinical characteristics, i.e., frequencies, and mean and standard deviations (*SD*). The existence of a significant association between two categorical variables (adherent/non-adherent) was assessed by the chi-squared test. Comparison of means among independent samples used Student’s *t*-test. Pearson’s correlation was used to test the existence of a linear association between two continuous variables, whereas Spearman’s correlation assessed the monotone association between two ordinal and noncontinuous variables.

Due to a marked skewed distribution, the PSDI obtained from BSI was dichotomized using an empirical cutoff value of 1.7 [34]. Logistic regression models were used to assess multiple correlations between each binary variable PST and PSDI and self-reported severity and discomfort, as well as the location of lesions. Multiple logistic regression models were used to assess the effect of the group formulations after adjusting for relevant variables as well as to investigate the impact of sociodemographic and psychological factors on therapy adherence. The level of significance was set at 0.05. The values of the size effect (phi-φ; *d* de Cohen), the chi-square test, and the *t*-test were calculated respectively. The statistical analyses were conducted using R 3.5.2 (R Computing, Vienna, Austria), a programming language and software environment for statistical computation.

## 3. Results

### 3.1. Patients’ Characteristics and Adherence Results

According to the SAPASI results, 35.6% (*n* = 36) of the patients presented mild psoriasis, 51.5% (*n* = 52) moderate, 10.9% (*n* = 11) a severe psoriasis condition, and 2 patients were in remission (2%). Adherence to topical treatment in psoriasis translated into positive associations between adherence and the education level (higher education) (*p* = 0.03; φ = 0.23), the single-family household (*p* = 0.01; φ = 0.44), active employment status (*p* = 0.049; φ = −0.19), familiar history of psoriasis (*p* = 0.04; φ = −0.21), and the presence of obsessive-compulsive symptoms (*p* = 0.01; *d* = 0.29). The size effect values are moderate for the variables of professional status, education, and family history of psoriasis and good for the number of family members (Table 1). The average adherence value obtained with the adherence combo was 65.4% ± 19.3%.

### 3.2. Sociodemographic Predictors of Adherence

The estimated values obtained with the multiple logistic regression model based on Equation (2):log(π/(1 − π)) = β_0 + β_1 HighEducation + β_2 SingleFamilyUnit + β_3 InactiveProfessionalStatus(2)

This model (Table 2) estimates that patients with low levels of education, who live in households with two or more family members and are in an inactive professional situation have the odds for adherence of exp (1.45) = 0.24. This means that the probability of their adhering to therapy is 76.4% lower than for patients with higher education, living in families with one member, and active professional status. Considering only the sociodemographic variables of the model, the best social situation for adherence is to live alone, to have an active professional situation, and to have a high education level.

### 3.3. Psychopathological Differences between Non-Adherence and Adherence Groups

Patients with obsessive-compulsive symptoms (*p* = 0.01; *d* = 0.29) adhered more to topical treatment of psoriasis (Table 3). Considering the other dimensions of psychopathology, there were no significant differences.

### 3.4. Psychopathological Predictors of Adherence

Being unable to find a model of psychopathological predictors based only on one dimension of psychopathology, i.e., obsession/compulsion, the model variables that were the most associated with obsession/compulsion were added, obtaining a model of psychopathological, clinical, and sociodemographic predictors of adherence. The estimated values obtained with the multiple logistic regression model were based on Equation (3) (Table 4):log(π/(1 − π)) = β_0 + β_1 Female + β_2 ObsessiveCompulsive + β_3 DiseaseDuration + β_5 Gender × DiseaseDuration(3)

There is a positive association between adherence and female gender and obsession/compulsion symptomatology. The odds of adherence for a male patient who does not suffer from obsessive-compulsive symptoms and who has been diagnosed with psoriasis for less than 1 year are estimated at exp (−2.46) = 0.09. It is expected that there is only an 8.6% probability of this patient not adhering to treatment. In the first years after the psoriasis diagnosis, the model predicts that women will adhere more to treatment than men and that for psoriasis diagnosed for a long time, this difference is no longer significant.

### 3.5. Psychosocial Predictors of Adherence

Finally, there is a psychosocial model combining the sociodemographic variables and psychopathological symptoms identified as predictors in the previous models. This model is translated in Equation (4) (Table 5):log(π/(1 − π)) = β_0 + β_1 HighEducation + β_2 SingleFamilyUnit + β_3 InactiveProfessionalStatus + β_5 ObssessiveCompulsive (4)

Positive associations between adherence and the high education level, family with one member, active professional situation, and obsessive-compulsive symptoms were estimated. Among these factors, it is expected that the factor with a higher impact on adherence is the single-family household, while the education level is the factor with the lowest impact. It is observed that the *p*-value of education is outside the limit set for the level of significance. However, this means that whenever assuming a nonzero effect for higher education, with a 6.1% error, only 1.1% above the 5% is tolerated.

## 4. Discussion

In this study, the influence of the psychosocial factors and clinical variables on psoriasis adherence to topical treatment was assessed. According to Carlsen, Olasz, Carlsen, and Serup [37], psoriasis patients adhere less to treatment due to their perceptions of psoriasis, medication choice, and personal factors. Soleymani, Reddy, Cohen, and Neimann [38] clarified that psoriasis patients’ non-adherence to treatments is due to fear and experience with adverse effects, therapy sessions, cost, poor instruction, poor communication between healthcare professionals and the patient, treatment regimens, incompatibility with patients’ daily activities, values, and beliefs. The sociodemographic or clinical factors (i.e., absence from work, social life impact, and disease duration) were not consistently associated with non-adherence to treatment of immune-mediated inflammatory diseases, and there was limited evidence of an association between non-adherence and treatment factors, such as dosing frequency [39]. However, adherence was associated with psychosocial factors, namely, healthcare professional–patient relationships, perceptions of treatment concerns and depression, lower treatment self-efficacy and necessity beliefs, and practical barriers to treatment. We found that sociodemographic factors such as education, household size, and family history of psoriasis had a statistically significant effect on treatment adherence. These results are not in agreement with the ones obtained by Svendsen, Möller, Feldman, and Andersen [40], who described that sociodemographic factors do not have a large influence on psoriasis adherence to topical treatment by patients. This difference in the results may be related to the fact that the methodologies for measurement adherence could be distinct [22]. Among the sociodemographic factors, the existence of a single-family household was the variable that most positively influenced adherence, while education was the factor with the lowest impact. One explanation for these results may be related to the fact that subjects who live alone have more time available to themselves to perform the daily application of topical treatments commonly used in psoriasis, often multiple times. In the literature, the influence of the household on adherence to treatment is rarely explored, while for the civil status, the results of different studies are contradictory [29]. However, marital status is not directly linked with the household dimension since single, divorced, separated, or widowed individuals may live in households with more than one member. The psychosocial model found a positive association between adherence and the level of education similar to that obtained by Gokdemir et al., 2008 [41]; however, Thorneloe et al. [29] indicated the absence of an association. The positive association between active professional status and adherence to treatment was also described by Zaghloul et al., 2004 [42]. One possible explanation is that people who are professionally active need to be at their best from an aesthetic point of view and are, thus, more committed to applying the treatment to minimize the appearance of lesions. A positive association of gender on adherence to topical treatment of psoriasis was found, similar to Colombo et al., 2014 [43], although in other cases, there was no gender effect on adherence [29]. Gender and duration of disease, variables with no statistically significant effect when analyzed individually, revealed an effect when included in the models, even with interaction. The literature on the impact of psychopathology on treatment adherence in psoriasis is scarce, an example being the study from Puig et al., 2015 [44], despite being abundant regarding the association between psychopathology and psoriasis, as patent in the studies by Ferreira et al., 2016 [31]. Obsessive-compulsive symptomatology is associated with perfectionism, need for control, and concern with the image, which may justify the positive effect on treatment adherence observed in this study. In patients in treatment with a psychologist or psychiatrist, issues related to the control of the disease are usually the target of intervention as well as the promotion of treatment adherence, which may be the basis of the positive association found [45]. The results of our study showed that the family history of psoriasis has a positive influence on treatment adherence since the group of patients with a family member with psoriasis adhered more to topical treatment. The effect of this variable has been sparsely studied, and in all studies, there has been an absence of association with treatment adherence [43]. However, relatives with the same disease can contribute to better knowledge about the pathology and therapeutics, particularly on the influence of adherence on treatment outcomes, thus leading to a more informed behavior toward treatment applications. This study used a new approach for adherence evaluation based on a multiple logistic regression model. The methodology to assess topical treatment adherence should rely on distinct measures since the results can vary significantly. A combined measure such as the presented adherence combo is thus recommended for future studies.

In concordance with our results that indicate the important role of the level of education on adherence, future research on evaluating the influence of interventions to improve health literacy on treatment adherence could be relevant—through an integrated approach with different health professionals (e.g., doctors, pharmacists, nurses, psychologists). Higher levels of education can be associated with higher scores of health literacy. Avazeh et al., 2020, found a positive correlation between health literacy and medication adherence and suggested improving access to the internet and information communication technologies and the development of patient education approaches and techniques aiming at the enhancement of treatment adherence [46].

One limitation of the study was a high experimental mortality (39.6%), which reduced the sample size. Most patients were recruited in hospital units (69.1%), which may have biased the results obtained. Regarding self-reported measures, the influence of social desirability on the results must be considered since the instruments were administered in the presence of a researcher, except for the medication diary (med log). To avoid this effect, the researcher did not explicitly inform patients that the objective of the study was to assess adherence. It should also be noted that in this study, another measure was used, in addition to self-reported measures, to assess adherence (medication weighing).

## 5. Conclusions

Psoriasis, a chronic inflammatory skin disease, significantly impacts patients’ quality of life across social, emotional, and professional dimensions, necessitating long-term treatment. This study investigated the influence of psychosocial and clinical factors on adherence to topical treatment in psoriasis patients. Self-reported measures and medication weighing to evaluate adherence, alongside psychopathological symptom assessments and a sociodemographic and clinical questionnaire, were employed. The results, based on a sample of 102 patients with psoriasis and logistic regression models, unveiled positive associations between adherence to topical treatment and higher education levels, single-family households, active employment status, familiar history of psoriasis, and the presence of obsessive-compulsive symptoms. These results underscore the importance of multidisciplinary treatment approaches that consider patients’ psychosocial profiles alongside clinical parameters. This study highlights the complex associations between psychosocial, clinical, and sociodemographic factors in adherence to topical treatment in psoriasis. By clarifying these associations, more targeted interventions can be implemented to increase adherence to treatment and improve the management of this chronic dermatological condition. Accordingly, this study contributes to filling gaps in understanding the multifaceted nature of treatment adherence in psoriasis, emphasizing the need for tailored interventions targeting specific patient characteristics. Some limitations of the study include the experimental mortality and the use of self-reported measures. To avoid this effect a new adherence measure was developed (adherence combo). Future research studies may explore the efficacy of interventions aimed at improving health literacy and addressing psychosocial barriers to adherence. Collaborative efforts involving healthcare professionals from multidisciplinary areas could enhance patient education and support, ultimately optimizing treatment outcomes for psoriasis patients.

## Figures and Tables

**Table 1 healthcare-12-00822-t001:** Sociodemographic and clinical characteristics of the sample and differences between non-adherence and adherence groups.

		Total Sample (*N* = 94; % = 100)	Non-Adherence (*N* = 71; % = 75.5)	Adherence (*N* = 23; % = 24.5)	*p*-value	Test	Size Effect
Age	Mean (*SD*)	49.4 (14.4)	49.3 (15.1)	49.7 (12.4)	0.90	*t*	*d*
	Min–Max	20–82				−0.17	−0.03
Gender	Male, *n* (%)	52 (55.3)	43 (82.7)	9 (17.3)	0.07	χ^2^	φ
	Female, *n* (%)	42 (44.7)	28 (66.7)	14 (33.3)		3.23	0.19
Marital status	Single/divorced/widowed, *n* (%)	31 (33.0)	20 (64.5)	11 (35.5)	0.08	χ^2^	φ
	Married, *n* (%)	63 (67.0)	51 (81.0)	12 (19.0)		0.30	−0.18
Education	Primary/secondary, *n* (%)	82 (87.2)	65 (79.3)	17 (20.7)	**0.03**	χ^2^	φ
	Higher education, *n* (%)	12 (12.8)	6 (50.0)	6 (50.0)		4.85	0.23
Professional status	Employed/student, *n* (%)	58 (61.7)	40 (69.0)	18 (31.0)	**0.05**	χ^2^	φ
	Unemployed/retired, *n* (%)	36 (38.3)	31 (86.1)	5 (13.9)		3.53	−0.19
Number of family members	1 person, *n* (%)	12 (12.8)	4 (33.3)	8 (66.7)	**0.01**	χ^2^	φ
	More than 1 person, *n* (%)	82 (87.2)	67 (81.7)	15 (18.3)		18.12	0.44
Family history of psoriasis	Yes, *n* (%)	52 (55.3)	35 (67.3)	17 (32.7)	**0.** **04**	χ^2^	φ
	No, *n* (%)	42 (44.7)	36 (85.7)	6 (14.3)		4.26	−0.21
Disease duration	Mean (*SD*)	19.5 (16.0)	19.6 (16.6)	19.0 (14.0)	0.86	*t*	*d*
	Min–Max	0–59	0–59	0–49		0.17	0.04
Changes in daily activities	Mean (*SD*)	3.2 (21.3)	8.1 (37.5)	1.62 (11.9)	0.43	*t*	*d*
	Min–Max	0–180	0–99	0–180		−0.81	−0.31
Absence from work	Mean (*SD*)	0.20 (0.95)	0.21 (1.05)	0.17 (0.58)	0.89	*t*	*d*
	Min–Max	0–180	0–99			0.14	0.04
Social life impact	Mild, *n* (%)	30 (31.9)	24 (80.0)	6 (20.0)		χ^2^	φ
	Moderate, *n* (%)	38 (40.4)	29 (76.3)	9 (23.7)	0.64	0.90	0.10
	Hight, *n* (%)	26 (27.7)	18 (69.2)	8 (30.8)			
Severity (SAPASI)	Mild, *n* (%)	23 (24.7)	16 (22.8)	7 (30.4)		χ^2^	φ
	Moderate, *n* (%)	59 (63.4)	45 (64.3)	14 (60.9)	0.71	0.69	0.09
	Severe, *n* (%)	11 (11.8)	9 (12.9)	2 (8.7)			
Psychotropic medication	Yes, *n* (%)	18 (19.1)	12 (66.7)	6 (33.3)	0.33	χ^2^	φ
	No, *n* (%)	76 (80.9)	59 (77.6)	17 (22.4)		0.95	−0.10

*N* = total sample; *n* = subsample; % = percentage; *SD* = standard deviation; **bold**: significant values.

**Table 2 healthcare-12-00822-t002:** Estimation from the logistic regression model for the sociodemographic predictors of adherence.

Variables	Estimates	*SD*	*p*-value	95% CI
Constant	−1.44	0.38	**<0.01**	−2.23; −0.76
Higher education	1.53	0.71	**0.03**	0.15; 2.91
Family with 1 member	2.96	0.85	**<0.01**	1.47; 4.81
Inactive professional status	−1.45	0.74	**0.03**	−3.14; −0.16

*SD* = standard deviation; CI = confidence interval; **bold**: significant values.

**Table 3 healthcare-12-00822-t003:** Psychosocial differences between non-adherence and adherence groups.

		Adherence Combo (*t*-test)					
		Non-Adherence		Adherence		*p*-value	*d*
		*n*	%	*n*	%		
Somatization	No	43	60.6	11	47.8	0.28	0.11
	Yes	28	39.4	12	52.2		
Obsession/Compulsion	No	60	84.5	13	56.5	**0.01**	**0.29**
	Yes	11	15.5	10	43.5		
Interpersonal sensitivity	No	54	76.1	16	69.6	0.54	0.06
	Yes	17	23.9	7	30.4		
Depression	No	54	76.1	16	69.6	0.54	0.06
	Yes	17	23.9	7	30.4		
Anxiety	No	47	66.2	13	56.5	0.40	0.09
	Yes	24	33.8	10	43.5		
Hostility	No	50	70.4	15	65.2	0.64	0.05
	Yes	21	29.6	8	34.8		
Phobic anxiety	No	53	74.6	14	60.9	0.20	0.13
	Yes	18	25.4	9	39.1		
Paranoid ideation	No	57	80.3	16	69.6	0.28	0.11
	Yes	14	19.7	7	30.4		
Psychoticism	No	66	93.0	18	78.3	0.06	0.21
	Yes	5	7.0	5	21.7		

*n* = subsample; % = percentage; χ^2^ = chi-squared distribution; *p* = *p*-value; **bold:** significant values.

**Table 4 healthcare-12-00822-t004:** Estimation of the model for psychopathological, clinical, and social predictors of adherence based on the logistic regression model.

Variables	Estimates	*SD*	*p*-value	95% CI
Constant	−2.46	0.69	**<−0.01**	−3.96; −1.27
Female gender	2.22	0.90	**0.01**	0.57; 4.07
Obsession/compulsion	1.67	0.60	**0.00**	0.54; 2.86
Disease duration	0.02	0.02	0.36	−0.03; 0.07
Disease duration ^a^	−0.07	0.04	**0.04**	−0.14; −0.00

*SD* = standard deviation; CI = confidence interval; **bold:** significant values; ^a^ only female gender.

**Table 5 healthcare-12-00822-t005:** Estimation of the model for the identification of psychosocial predictors of adherence based on the logistic regression model.

Variables	Estimates	*SD*	*p*-value	95% CI
Constant	−1.80	0.44	**<0.01**	−2.73; −1.02
Higher education	1.36	0.74	0.061	−0.07; 2.77
Family with 1 member	3.00	0.92	**<0.01**	1.40; 5.04
Inactive professional status	−1.49	0.76	**0.03**	−3.21; −0.16
Obsession/compulsion	1.49	0.64	**0.02**	0.27; 2.78

*SD* = standard deviation; *p* = *p*-value; CI = confidence interval; **bold:** significant values.

## Data Availability

Data will be made available on request.

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
