# Peer review of "Influence of Clinical and Psychosocial Factors on the Adherence to Topical Treatment in Psoriasis"

_healthcare, 2024, doi:10.3390/healthcare12080822_

Round 1

Reviewer 1 Report

Comments and Suggestions for Authors

1.     The p values in the abstract don’t tell the reader the magnitude of the relationship.  Give the magnitude, not just the p value.

2.     I cannot tell from the abstract if any of the abstract conclusions are justified.  It may be that adherence should be addressed in all patients, not just ones that have some modest tendency toward lower adherence.  I didn’t see any evidence presented that counseling would improve the adherence.

3.     The abstract should state the major limitation.

4.     “a high variability of adherence results was found” can be written more concisely as “Adherence was highly variable.”  Never use words like known, shown, proven, reported or demonstrated.  If something is any of those, it just is.

5.     “are able to” is a long way to say “can”

6.     “Literature has demonstrated that” is a verbiage that should be deleted.  Phrases ending in “that” are useless verbiage that should be deleted.

7.     “Being important to consider the biopsy-chosocial dimension” might be less confusing as “Being important to consider the bio-psychosocial dimension”

8.     “Positive association with adherence has been reported for patient satisfied with treatment and a negative relationship with adherence for psychological distress” would be more concisely written as “Adherence is positively associated with treatment satisfaction and negatively associated with psychological distress.”

9.     To refer to a table or figure, just put "(Table X)" or "(Figure x)" at the end of an appropriate sentence.

10.  The p values in the Results don’t tell the reader the magnitude of the relationship.  Give the magnitude, not just the p value.

11.  With 100 subjects, reporting values with 4 significant digits of precision is inappropriate.  Only 2 digits of precision is probably appropriate throughout.  Using too many digits misleads the reader into thinking the numbers are more precise than they are.

12.  The table is very difficult to interpret.  Let’s say I want to understand the impact of higher education on adherence.  Of the patients with higher education, 79% were non adherent and 21% were adherent.  But there were 3 times as many non-adherent than adherent patients.  So a lot of the difference is simply due to the higher % of non-adherent patients in the study group.  Can you find a way to present the data that makes it easier to understand the impact of different variables on adherence?

13.   I find “The model, presented in table 2 estimates that patients with low levels of education, who live in households with 2 or more elements and are in an inactive professional situation have an odds for adherence of exp (1.445) = 0.236. This means that the probability of adhering to therapy is 76.4% lower than the probability of non-adhering” uninterpretable.  This doesn’t tell me how this group of people’s adherence compares to others.

14.  “In a systematic review carried out by 278 Vangeli and colleagues [31], about factors associated with treatment non-adherence for 279 immune-mediated inflammatory diseases, it was found that” is a lot of useless verbiage.

15.  “the same study found an association between 283 adherence and psychosocial factors” is a long way to say “Adherence was associated with psychosocial factors.”

16.  “The results obtained in this study 286 revealed that” is also verbiage.

17.  I don’t see the logic to support “The results obtained in this study 286 revealed that sociodemographic factors such as education, household and family history 287 of psoriasis, had a statistically significant effect on treatment adherence. These results are 288 not in agreement with the ones obtained by Svendsen, Möller, Feldman, and Andersen 289 [32], who reported that sociodemographic factors do not have a large influence on psori- 290 asis adherence to topical treatment by patients.”  One study found a “statistically significant effect” but that doesn’t imply that there is a large influence.  Statistical significance does not imply a large magnitude.  A major weakness of this study is that it doesn’t clearly present what the magnitude is.

18.  Also, a strength of Svendsen’s work is the use of objective electronic monitors.  Self report may not be accurate, and psychosocial factors may be affecting how inaccurate the reporting is, not necessarily the adherence behavior being measured.

19.  Everything should be noted.  “It should be noted that” is verbiage.

20.  The Discussion is overly long with a lot of unsupported speculation that can be deleted.

21.  Never claim to be first.  It is not scientifically relevant, and there's no way to prove it would be true at the time of publication.

22.  The Discussion lacks discuss of limitations.

Comments on the Quality of English Language

Good enough

Reviewer 2 Report

Comments and Suggestions for Authors

In the Discussion paragraph, in the sentences in which you present your results, emphasize that these are the results of your research. line 292, 302, 308,328
For example: The results of our study showed...
or We found that...

Reviewer 3 Report

Comments and Suggestions for Authors

The authors have presented the study about clinical and psychosocial factors on the adherence to topical treatment in psoriasis .The overall scope is well met and the methods are appropriately chosen. The introduction is concise and informative, but it could include more information about results of previous studies about topical treatment adherence (as well as in discussion). The results are important as a matter of the topic. However, there are a few concerns that can be appropriately addressed to improve the paper. My remarks are the following: 

  1. References related to psoriasis should be more updated.
  2. The material and methods didn’t mention the type of topical treatment and formula (and also how formula impact the adherence) or location of the skin lesions (if it’s eg scalp or nail psoriasis), which are important as a matter of adherence. 
  3. Line 171 - the time of applications is 45 days, which was considered by the experts’ panel, composed by dermatologists, pharmacists, psychologists and statistics  - could you include more details of that decision 
  4. Lines 201-203: there is just SAPASI results (just in % no N, N should be added). Did you measure PASI or BSA? The difference in severity of the skin lesion between adherence and not adherence should be calculated in place in table 1. 
  5. In title authors mentioned clinical factor. In my opinion study is focused on psychosocial not clinical factors. 
  6. Lines 288-291 - the reasons/ possible explanations for disagreement from previous study should be added 
  7. There is lack informations about limitations of the study in discussion. 

Reviewer 4 Report

Comments and Suggestions for Authors

This is an interesting work on patients'adherence on topical treatments for plaque psoriasi.

I have a few suggestions to further improve the manuscript:

In the materials and methods section, the Authors should describe more deeply the Scores and Questionairres that they have used, since most clinicians may not know well those scores. 

Also, They should explain better how the Adherence Combo was calculated (page 4, line 155)

Also, I would deeply discuss why the Authors believe that many factors (disease duration, social life impact, absence from work) did not impact the adherence

Comments on the Quality of English Language

English language is fine, only some minor revisions are required

Round 2

Reviewer 4 Report

Comments and Suggestions for Authors

The manuscript has been greatly improved.

I have no further comments